# Prevalence and Risk Factor Analysis of Post-Intensive Care Syndrome in Patients with COVID-19 Requiring Mechanical Ventilation: A Multicenter Prospective Observational Study

**DOI:** 10.3390/jcm11195758

**Published:** 2022-09-28

**Authors:** Junji Hatakeyama, Shigeaki Inoue, Keibun Liu, Kazuma Yamakawa, Takeshi Nishida, Shinichiro Ohshimo, Satoru Hashimoto, Naoki Kanda, Shuhei Maruyama, Yoshitaka Ogata, Daisuke Kawakami, Hiroaki Shimizu, Katsura Hayakawa, Aiko Tanaka, Taku Oshima, Tatsuya Fuchigami, Hironori Yawata, Kyoji Oe, Akira Kawauchi, Hidehiro Yamagata, Masahiro Harada, Yuichi Sato, Tomoyuki Nakamura, Kei Sugiki, Takahiro Hakozaki, Satoru Beppu, Masaki Anraku, Noboru Kato, Tomomi Iwashita, Hiroshi Kamijo, Yuichiro Kitagawa, Michio Nagashima, Hirona Nishimaki, Kentaro Tokuda, Osamu Nishida, Kensuke Nakamura

**Affiliations:** 1Department of Emergency and Critical Care Medicine, National Hospital Organization Tokyo Medical Center, Tokyo 152-8902, Japan; 2Department of Emergency and Critical Care Medicine, Osaka Medical and Pharmaceutical University, Osaka 569-8686, Japan; 3Department of Disaster and Emergency Medicine, School of Medicine, Kobe University, Kobe 650-0017, Japan; 4Critical Care Research Group, The Prince Charles Hospital, Brisbane, QLD 4032, Australia; 5Division of Trauma and Surgical Critical Care, Osaka General Medical Center, Osaka 558-8558, Japan; 6Department of Emergency and Critical Care Medicine, Graduate School of Biomedical and Health Sciences, Hiroshima University, Hiroshima 734-8551, Japan; 7Department of Intensive Care Medicine, Kyoto Prefectural University of Medicine, Kyoto 602-8566, Japan; 8Division of General Internal Medicine, Jichi Medical University Hospital, Tochigi 329-0498, Japan; 9Department of Emergency and Critical Care Medicine, Kansai Medical University Medical Center, Osaka 570-8507, Japan; 10Department of Critical Care Medicine, Yao Tokushukai General Hospital, Osaka 581-0011, Japan; 11Department of Anesthesia and Critical Care, Kobe City Medical Center General Hospital, Kobe 650-0047, Japan; 12Acute Care Medical Center, Hyogo Prefectural Kakogawa Medical Center, Kakogawa 675-8555, Japan; 13Department of Emergency and Critical Care Medicine, Saitama Red Cross Hospital, Saitama 330-8553, Japan; 14Department of Anesthesiology and Intensive Care Medicine, Osaka University Graduate School of Medicine, Osaka 565-0871, Japan; 15Department of Emergency and Critical Care Medicine, Chiba University Graduate School of Medicine, Chiba 260-8677, Japan; 16Intensive Care Unit, University of the Ryukyus Hospital, Nishihara 903-0215, Japan; 17Department of Emergency and Critical Care Medicine, Japanese Red Cross Kyoto Daiichi Hospital, Kyoto 605-0981, Japan; 18Department of Intensive Care Medicine, Asahi General Hospital, Chiba 289-2511, Japan; 19Japanese Red Cross Maebashi Hospital, Advanced Medical Emergency Department and Critical Care Center, Maebashi 371-0811, Japan; 20Advanced Critical Care and Emergency Center, Yokohama City University Medical Center, Yokohama 232-0024, Japan; 21Department of Emergency and Critical Care, National Hospital Organization Kumamoto Medical Center, Kumamoto 860-0008, Japan; 22Critical Care and Emergency Center, Metropolitan Tama General Medical Center, Tokyo 183-8524, Japan; 23Department of Anesthesiology and Critical Care Medicine, Fujita Health University School of Medicine, Nagoya 470-1192, Japan; 24Department of Intensive Care Medicine, Yokohama City Minato Red Cross Hospital, Yokohama 231-8682, Japan; 25Department of Anesthesiology, Fukushima Medical University, Fukushima 960-1295, Japan; 26Department of Emergency and Critical Care Medicine, National Hospital Organization Kyoto Medical Center, Kyoto 612-8555, Japan; 27Department of Thoracic Surgery, Tokyo Metropolitan Geriatric Hospital and Institute of Gerontology, Tokyo 173-0015, Japan; 28Department of Emergency and Critical Care Medicine, Yodogawa Christian Hospital, Osaka 533-0024, Japan; 29Department of Emergency and Critical Care Center, Nagano Red Cross Hospital, Nagano 380-8582, Japan; 30Department of Emergency and Critical Care Medicine, Shinshu University Hospital, Nagano 390-8621, Japan; 31Emergency and Disaster Medicine, Gifu University School of Medicine Graduate School of Medicine, Gifu 501-1194, Japan; 32Department of Intensive Care Medicine, Tokyo Medical and Dental University, Tokyo 113-8519, Japan; 33Department of Anesthesiology and Pain Medicine, Juntendo University, Tokyo 113-8431, Japan; 34Department of Anesthesiology, Tohoku University Hospital, Sendai 980-8574, Japan; 35Intensive Care Unit, Kyushu University Hospital, Fukuoka 812-8582, Japan; 36Department of Emergency and Critical Care Medicine, Hitachi General Hospital, Hitachi 317-0077, Japan

**Keywords:** COVID-19, critically ill patients, delirium, post-intensive care syndrome

## Abstract

Introduction: Post-intensive care syndrome (PICS) is an emerging problem in critically ill patients and the prevalence and risk factors are unclear in patients with severe coronavirus disease 2019 (COVID-19). This multicenter prospective observational study aimed to investigate the prevalence and risk factors of PICS in ventilated patients with COVID-19 after ICU discharge. Methods: Questionnaires were administered twice in surviving patients with COVID-19 who had required mechanical ventilation, concerning Barthel Index, Short-Memory Questionnaire, and Hospital Anxiety and Depression Scale scores. The risk factors for PICS were examined using a multivariate logistic regression analysis. Results: The first and second PICS surveys were obtained at 5.5 and 13.5 months (mean) after ICU discharge, with 251 and 209 patients completing the questionnaires and with a prevalence of PICS of 58.6% and 60.8%, respectively, along with the highest percentages of cognitive impairment. Delirium (with an odds ratio of (OR) 2.34, 95% CI 1.1–4.9, and *p* = 0.03) and the duration of mechanical ventilation (with an OR of 1.29, 95% CI 1.05–1.58, and *p* = 0.02) were independently identified as the risk factors for PICS in the first PICS survey. Conclusion: Approximately 60% of the ventilated patients with COVID-19 experienced persistent PICS, especially delirium, and required longer mechanical ventilation.

## 1. Introduction

Functional disabilities that occur during an intensive care unit (ICU) stay or after an ICU or hospital discharge include physical, cognitive, and mental impairments, also known as post-intensive care syndrome (PICS), with effects on the long-term prognosis of patients who survive an ICU discharge [1]. It has been reported that 50–70% of patients admitted to ICU develop PICS [2], and that the prevalence of PICS leads to a decrease in the quality of life (QOL) and a disengagement from work, which have a significant effect on society [3,4,5]. For example, the survival rate of patients with sepsis has certainly improved [6]; however, approximately one-third of patients with sepsis continue to have some difficulties in their lives after leaving the ICU, and the long-term prognosis of patients with sepsis has not yet improved [7].

Coronavirus disease 2019 (COVID-19) is an infectious disease caused by severe acute respiratory syndrome coronavirus 2 (SARS-CoV-2) with a reported high mortality rate once mechanical ventilation is required [8,9,10]. Various long-term sequelae of general COVID-19 infection often have been reported, known as long-COVID or post-acute COVID-19, such as pulmonary sequelae, neurologic disorders, difficulty concentrating, fatigue, and muscle weakness [11,12,13]. PICS after a severe COVID-19 infection is likely to differ from other ICU-related diseases in terms of pulmonary sequelae and neurologic disorders, and can possibly disrupt daily life in different ways [14,15,16,17]. While some studies have investigated the prevalence of PICS after COVID-19 [14,15,18,19,20], there have been few studies primarily comprised of ventilated patients and of an adequate sample size in which all the components of PICS in terms of physical, cognitive, and mental impairments were evaluated simultaneously. Thus, it remains necessary to clarify the prevalence and risk factors of PICS in patients with severe COVID-19 to improve their long-term prognoses.

We conducted a large-scale clinical study to investigate and clarify the prevalence and risk factors of PICS in critically ill patients with COVID-19 who required mechanical ventilation during hospitalization.

## 2. Materials and Methods

### 2.1. Study Design and Setting

This study forms part of a multicenter observational study, “Post-intensive care outcomes in patients with Coronavirus Disease 2019 study” (PICS-COVID study) that was conducted in collaboration with the Cross ICU Searchable Information System (CRISIS), the national registry in Japan for ICU patients with COVID-19 who require mechanical ventilation or extracorporeal membrane oxygenation (ECMO), covering 80% of the ICU beds throughout Japan [21].

All the patients with COVID-19 admitted to 32 ICUs were considered for this study. A central office was established for the performance of administrative tasks, which included mailing questionnaires to the patients, collecting and tabulating the responses in the questionnaires, and handling inquiries from patients. Details of the participating facilities and central office are outlined in the Appendix A.

Approval for this study was granted by the Institutional Review Board of the National Hospital Organization Tokyo Medical Center (date: 26 November 2020, approval number: R20-133) and the review boards of each participating hospital. The study protocol was registered in the University Hospital Medical Information Network (UMIN000041276).

### 2.2. Study Population and Eligibility Criteria

The PICS survey was conducted among patients discharged from the ICU between March and December 2020. The inclusion criteria for this study included patients with COVID-19 aged ≥ 20 years who required invasive mechanical ventilation during hospitalization. The indications for invasive mechanical ventilation management were determined at the discretion of the participating institutions. A SARS-CoV-2 infection was confirmed using a polymerase chain reaction test. The exclusion criteria comprised patients from whom written informed consent could not be obtained; and patients who were unable to walk on their own before admission, regardless of the use of assistive devices, because of the possibility of pre-existing PICS. All patients with COVID-19 who require mechanical ventilation are promptly registered in the CRISIS registry in accordance with the national policy when they are admitted to the ICUs of each participating institution. The patients registered in the CRISIS registry were enrolled in our study if they met the inclusion criteria. Written informed consent was obtained from all patients in the analysis.

### 2.3. Procedures

Two surveys were conducted to evaluate PICS, with questionnaires sent to patients in February and October 2021. We did not send questionnaires to those who could not be contacted post-discharge. The central office made a phone call prior to sending the first PICS survey. Patients who did not respond to the first questionnaire were not sent a second questionnaire. In the second PICS survey, a phone call was also made prior to sending the questionnaire. The questionnaire consisted of simple questions regarding PICS. The Barthel Index (BI) [22,23], Short-Memory Questionnaire (SMQ) [24], Hospital Anxiety and Depression Scale (HADS)-anxiety, HADS-depression [25], and EQ-5D-5L [26] were used for the assessment of physical function, cognitive function, mental health, and QOL, respectively. Responses provided from someone approved by the patient to act in their place were permitted. All the questionnaire items are listed in the Appendix A. The responses were collected and tabulated at the central office. The patients who responded to the survey were given an incentive worth 1000 yen per survey.

### 2.4. Variables and Measurements

All the variables and measurements are listed in the Appendix A.

### 2.5. Outcomes

The primary outcome was the prevalence of PICS after an ICU discharge as shown in the first PICS survey. The secondary outcomes were the prevalence of PICS after an ICU discharge as shown in the second PICS survey and the prevalence of three elements in the PICS and answers to all the other questions (see the Appendix A). The risk factors were analyzed for PICS in terms of the presence of three components, namely, a physical impairment, cognitive impairment, or mental disorder [1]. In this study, PICS was defined as any one of the following functional impairments: a physical impairment was defined as a score of ≤90 points on the BI [27], cognitive impairment as a score of <40 points on the SMQ [28], and mental disorder as a score of ≥8 points on the HADS-anxiety or HADS-depression scale [29,30].

### 2.6. Statistical Analysis

The demographic characteristics and long-term health outcomes of the patients were presented as medians (interquartile ranges) for continuous variables and as absolute values and percentages for categorical variables. The continuous variables of patient characteristics were compared using a Mann–Whitney U test, and the categorical variables were compared using a chi-squared test. The risk factors for PICS (physical and cognitive impairment, and mental disorder) were analyzed using a multiple logistic regression analysis in relation to the following clinically relevant variables: age (per 10 years increase) [31], male sex [31,32], body mass index (BMI) (per 5 kg/m^2^ increase) [33,34], the Clinical Frailty Scale [35], sequential organ failure assessment (SOFA) [32], delirium [36], duration of mechanical ventilation (per 7 days increase) [32], ECMO [37], continuous neuromuscular blocking drugs [32], maximum daily dose of prednisolone equivalent (per 50 mg/day increase) [38], prone positioning [39], rehabilitation program in the ICU conducted by physical therapists [40], and the time period. The time period refers to the period from ICU survival discharge to February 2021 for the first PICS survey and from ICU survival discharge to October 2021 for the second PICS survey. A univariate regression analysis using a restricted cubic spline model was used to analyze the nonlinear relationship between the three PICS components and age [41,42]/BMI [33,34]. An estimated probability for the three PICS components with 95% confidence intervals (CIs) were calculated and depicted using spline curves. When missing values were noted in a patient’s questionnaire responses, the nominal scale was analyzed as zero, and the continuous variables were excluded from the analysis. A *p*-value < 0.05 (two-sided) was considered statistically significant. The spline curves were analyzed using R language, version 4.1.1 (R Foundation for Statistical Computing, Vienna, Austria) software; all the other data were analyzed using SPSS software, version 22 (IBM, Chicago, IL, USA).

## 3. Results

The study outline is shown in Figure 1. During the study period, 562 patients were treated with mechanical ventilation, and 410 eligible patients were registered in the study. Seventy-six patients died in hospital and 334 patients were discharged alive. A total of 251 surviving patients completed the questionnaire in the first PICS survey, and 209 completed the questionnaire in the second PICS survey. The survey response rates were 75.1% and 83.3%, respectively. All those who responded to the second PICS survey had also responded to the first PICS survey. None of the patients had missing clinical information; however, some questionnaire responses had missing data (Appendix A).

Patient characteristics stratified according to the PICS prevalence in each survey are shown in Table 1 and Table 2. No patient had already been admitted to ICU prior to their enrollment in this study. The first and second surveys were evaluated at a mean of 5.5 and 13.5 months after ICU discharge, respectively. The first PICS survey showed significant differences in the age, BMI, delirium, duration of mechanical ventilation, length of ICU and hospital stay, reintubation, and prednisolone dose with and without PICS, but there were no differences in these patient backgrounds except for the BMI in the second survey. In both the first and second PICS surveys, a lower BMI was associated with a significantly higher prevalence of PICS. The patient backgrounds of those who dropped out of the first and second surveys did not differ from the patient backgrounds of those for whom PICS could be evaluated. The patient background and PICS outcomes for patients who responded to both the first and second questionnaires are shown in the Appendix A. The patient background of the 42 dropout patients before the second questionnaire is shown in the Appendix A.

The prevalence of PICS among the patients with COVID-19 who required ventilatory management during hospitalization and the EQ-5D-5L values of each patient in the first and second surveys are shown in Figure 2 and Figure 3. PICS was diagnosed in 147 patients (58.6%) in the first survey and 127 patients (60.8%) in the second survey. There was no difference found in the percentage of those with a functional disability between the first and the second survey, with the most common functional disability being a cognitive impairment in 117 (46.6%) patients in the first survey and 111 (53.1%) patients in the second survey. The second most common functional disability was a mental disorder, identified in 80 (31.9%) patients who responded to the first survey and 60 (28.7%) patients in the second survey, while a physical impairment was identified in 55 (21.9%) patients in the first survey and 39 (18.7%) patients in the second survey. Two or more functional disabilities were present at the same time in 80 (31.9%) patients in the first survey and 66 (31.6%) in the second survey. The EQ-5D-5L values, which indicate the QOL, tended to be lower when two types of functional disability occurred simultaneously, and they were lowest among patients with all three types of functional disability.

Table 3 shows the responses to the questionnaire, categorized according to the survey, of those with and without PICS. Of the 251 patients in the first PICS survey with PICS, the median BI score was 100 (95–100), the median SMQ score was 40 (34.8–44), the median HADS score was 8 (3.8–14), and the median EQ-5D-5L score was 0.831 (0.710–1). The most common subjective symptom was weight loss, followed by anxiety, executive dysfunction, and dyspnea. Of the 209 patients in the second PICS survey with PICS, the median BI score was 100 (95–100), the median SMQ score was 39 (34–43), the median HADS score was 7 (3–14), and the median EQ-5D-5L score was 0.844 (0.759–1). The most common subjective symptom was anxiety, followed by dyspnea, executive dysfunction, and sleep disorder.

The results of the multiple logistic regression analysis are shown in a forest plot. In terms of PICS, delirium (with an odds ratio [OR] of 2.34, 95% CI 1.1–4.9, and *p* = 0.03) and the duration of mechanical ventilation per 7 days increase (with an OR of 1.29, 95% CI 1.05–1.58, and *p* = 0.02) were independent risk factors for PICS in the first PICS survey; however, in the second PICS survey, none of these factors were found to be associated with the prevalence of PICS (Figure 4). In the first PICS survey, the details concerning the three components of PICS were as follows (Figure 5): for cognitive impairment, a duration of mechanical ventilation increase was an independent risk factor; for mental disorder, delirium and a duration of mechanical ventilation increase were independent risk factors; and for physical impairment, an age increase, being female, delirium, and the duration of mechanical ventilation were independent risk factors. This trend was similar in the second PICS survey, with the exception of cognitive impairment (Figure 5).

To analyze the nonlinear risk for the prevalence of each component of PICS, univariate spline curves were used to examine the association between age/BMI and the three PICS components. An older age tended to be more likely to cause a cognitive and physical impairment, but was less likely to be associated with mental disorder. A low BMI was more likely to be related to a functional impairment in all three PICS factors, whereas obesity was less likely to be related to functional impairment. The first and second surveys showed similar spline curves (Figure 6).

## 4. Discussion

In patients with COVID-19 requiring mechanical ventilation, the prevalence of PICS after a mean ICU discharge of 5.5 and 13.5 months was 58.6% and 60.8%, respectively, with cognitive impairment being the most common type of functional impairment. Delirium and the duration of mechanical ventilation were independent risk factors for the prevalence of PICS.

In terms of the prevalence of PICS in general ICU patients, the prevalence has been reported as 64% and 56% at three and twelve months after ICU discharge; at three months, the cognitive impairment was 37.6%, mental impairment was 33%, and physical impairment was 23.6%; at twelve months, the cognitive impairment was 32.6%, mental impairment was 30.9%, physical impairment was 17.5%, while the prevalence of multiple functional impairments was 25% at three months and 21% at twelve months [43]. In an epidemiological study of PICS in ventilated patients in Japan, the prevalence of PICS after six months from ICU discharge was found to be 63.5%, with 37.5% having cognitive impairment, 14.6% having mental impairment, 32.3% having physical impairment, and 17.8% having multiple functional impairments [44]. Compared to previous studies, the present study showed no difference in the prevalence of PICS; however, a novel finding of this study was that patients with COVID-19 requiring mechanical ventilation had an unusually high prevalence of cognitive impairment. SARS-CoV-2 directly invades the lung tissue, causing a systemic inflammatory response and microvascular damage, leading to cerebral neuropathy [45,46] and affecting cognitive functions [47]. Moreover, the administration of corticosteroids during the acute phase of COVID-19 may cause adverse effects in the central nervous system, such as cognitive impairment, sleep disorders, and delirium [48]. Thus, PICS, in terms of cognitive dysfunction, may be more common after a severe COVID-19 infection.

Furthermore, delirium was found to be strongly associated with the prevalence of PICS in this study. Delirium is generally considered to be a risk factor for cognitive dysfunction [36]. Delirium in critically ill patients with COVID-19 is likely due to microvascular disease and inflammatory mechanisms [49]. Another key mechanism of delirium in patients with COVID-19 is the development of secondary encephalopathy, possibly related to the cytokine storm phenomenon. Immune-mediated injury is mainly due to cytokines and the activation of T lymphocytes, macrophages, and endothelial cells [50]. Furthermore, these cytokines may damage the blood–brain barrier and induce encephalopathy [50]. The development of delirium in patients with COVID-19 might be influenced by patient factors such as age, sex, severity of illness, and comorbidities, as well as environmental factors such as a poor compliance with the ABCDEF bundle in relation to patients being managed for COVID-19 [51], medications such as midazolam and steroids, and visitation restrictions that may also contribute to the prevalence of delirium [49,52]. It is unclear whether bundling responses improve the incidence and outcome of delirium in patients with COVID-19, but if delirium is a strong risk factor for PICS, as shown in this study, it may be necessary to make efforts to minimize delirium through providing standard ICU care to patients with COVID-19 [53].

Another risk factor for the prevalence of PICS found in this study was the duration of mechanical ventilation. In patients with COVID-19, a prolonged duration of mechanical ventilation, along with prolonged immobilization, can lead to prolonged inflammation followed by immunodeficiency and hypercatabolism. These conditions are referred to as persistent inflammation, immune suppression, and catabolism syndrome (PIICS) [54]. A PIICS prevalence can lead to persistent mild inflammation, the coexistence of anti-inflammation and immunosuppression to counteract or further counteract inflammation and cause infection, and a prolonged wasting of the body, especially the muscles, which can lead to the development of PICS [54,55,56]. Appropriate nutrition therapy and rehabilitation have been promoted as possible PIICS treatment interventions [57], and PIICS may also be controlled by providing appropriate ICU care and interventions intended to prevent PICS [56].

This study identified specific trends in relation to age/BMI and the three PICS components in patients with COVID-19. The prevalence of physical and cognitive impairment tended to be higher in older as well as in general critically ill patients [58,59]. The prevalence of mental disorder in general critically ill patients has been shown to exhibit a bimodal pattern between younger and older patients [60,61]. However, there was no age-related trend in the prevalence of mental disorders found in this study. This may be related to the relatively high prevalence of anxiety and depression among working-age adults [62]. Obesity is known to be associated with increased mortality in patients with COVID-19 [63,64], but was not found to be associated with the prevalence of functional disability in this study; rather, a low BMI was associated with worse trends for functional disability. The mechanism by which being underweight confers health risks has been suggested to be due to hypoleptinemic disorders characterized as fat loss, severe insulin resistance, hypertriglyceridemia, and ectopic fat accumulation [65,66]. In older adults with frailty, not only physical frailty but also cognitive impairment and mental disorder have been associated with frailty, and frailty is an independent risk factor for delirium, moderate cognitive impairment, and dementia [67,68].

This study had some limitations. First, since patients requiring ventilatory management were included in this study, it was difficult to assess the cognitive function and mental health in the acute phase. Only patients who could walk unassisted were selected; therefore, their physical and cognitive functions were stabilized to some extent before hospitalization. However, some patients might have had certain organic mental disorder characteristics or mild cognitive dysfunction prior to the onset of COVID-19. Second, the time taken to assess PICS and QOL differs between patients, which might have affected the internal validity of the results. Third, this study only involved the assessment of outcomes obtained using self-reported measures, which might have caused data bias. Fourth, PICS studies conducted using different assessment tools may not be comparable, and a minor physical impairment may have been missed using the BI. Fifth, we could not compare the prevalence and risk factor of PICS between ventilated- and non-ventilated patients with COVID-19 as non-ventilated patients with COVD-19 were not included in this study. Further research is needed in which these limitations are addressed.

## 5. Conclusions

Approximately 60% of the ventilated patients with COVID-19 were found to have experienced persistent PICS after 5.5 and 13.5 months of ICU discharge, with a higher prevalence of cognitive impairment than physical impairment and mental disorder. Delirium and a longer mechanical ventilation were identified as independent risk factors for PICS after 5.5 months of ICU discharge.

## Figures and Tables

**Figure 1 jcm-11-05758-f001:**
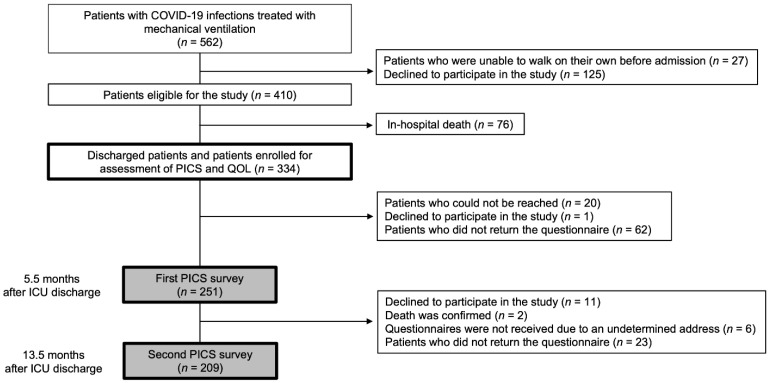
Study outline. Flowchart depicting the enrollment of subjects in the study. COVID-19: coronavirus disease 2019; ICU: intensive care unit; PICS: post-intensive care syndrome; QOL: quality of life.

**Figure 2 jcm-11-05758-f002:**
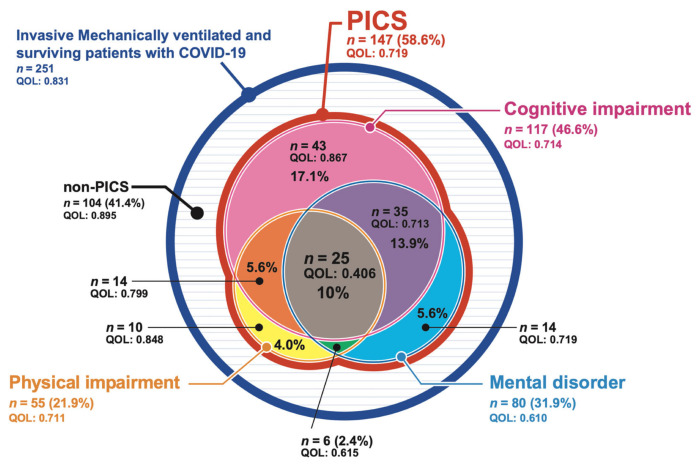
The prevalence of post-intensive care syndrome (PICS) after intensive care unit (ICU) discharge on the first PICS survey. The details of the prevalence of PICS, as well as the EQ-5D-5L values for each, among patients with coronavirus disease 2019 (COVID-19) who required ventilatory management during admission. EQ-5D-5L values are expressed as quality of life. PICS: post-intensive care syndrome; QOL: quality of life.

**Figure 3 jcm-11-05758-f003:**
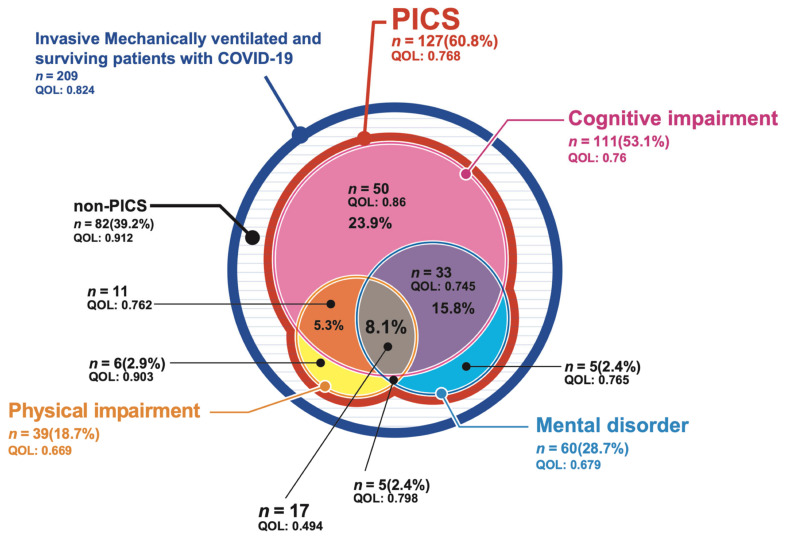
The prevalence of post-intensive care syndrome (PICS) after intensive care unit (ICU) discharge on the second PICS survey. The details of the prevalence of PICS, as well as the EQ-5D-5L values for each, among patients with coronavirus disease 2019 (COVID-19) who required ventilatory management during admission. EQ-5D-5L values are expressed as quality of life. PICS: post-intensive care syndrome; QOL: quality of life.

**Figure 4 jcm-11-05758-f004:**
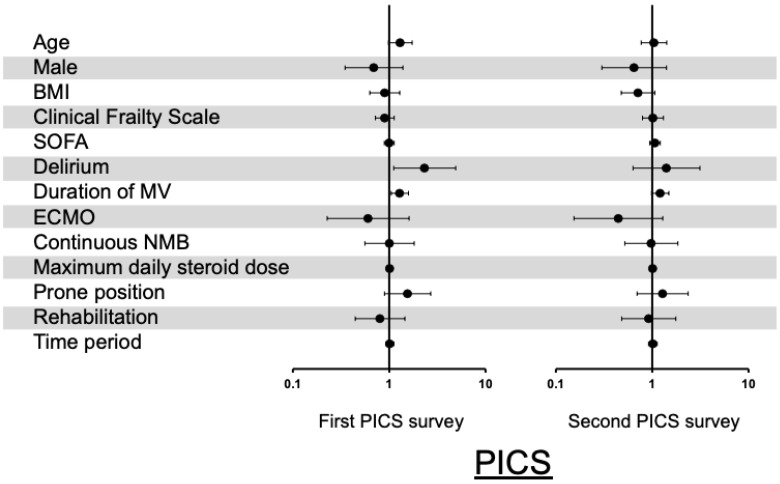
Risk factors for post-intensive care syndrome (PICS). Results of multiple logistic regression analysis of PICS are shown in a forest plot. Age was calculated in increments of 10 years, body mass index in increments of 5 kg/m^2^, mechanical ventilation period in increments of 7 days, and steroid dose in increments of 50 mg/day. Time period refers to the period from ICU survival discharge to February 2021 for the first PICS survey and from ICU survival discharge to October 2021 for the second PICS survey. BMI: body mass index; ECMO: extracorporeal membrane oxygenation; MV: mechanical ventilation; NMB: neuromuscular blocker; PICS: post-intensive care syndrome; SOFA: sequential organ failure assessment.

**Figure 5 jcm-11-05758-f005:**
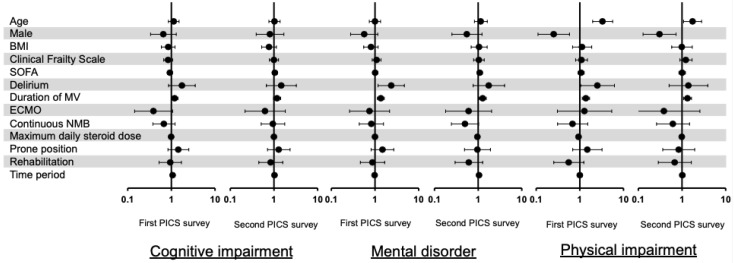
Risk factors for three elements of PICS. The results of multiple logistic regression analysis of cognitive impairment, mental disorder, and physical impairment are shown in a forest plot. Age was calculated in increments of 10 years, body mass index (BMI) in increments of 5 kg/m^2^, mechanical ventilation period in increments of 7 days, and steroid dose in increments of 50 mg/day. Time period refers to the period from intensive care unit (ICU) survival discharge to February 2021 for the first post-intensive care syndrome (PICS) survey, and from ICU survival discharge to October 2021 for the second PICS survey. BMI: body mass index; ECMO: extracorporeal membrane oxygenation; MV: mechanical ventilation; NMB: neuromuscular blocker; PICS: post-intensive care syndrome; SOFA: sequential organ failure assessment.

**Figure 6 jcm-11-05758-f006:**
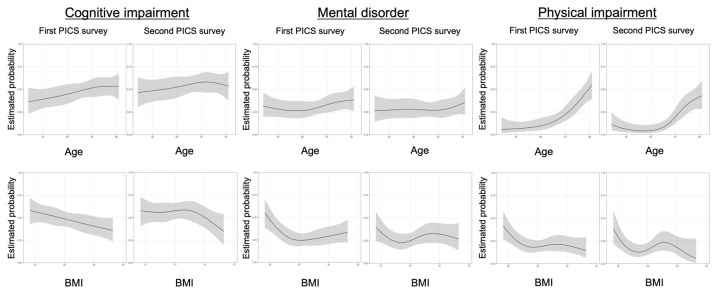
The relationship between three elements of post-intensive care syndrome (PICS) and age/body mass index by spline curves. The relationship between cognitive impairment/mental disorder/physical impairment and age/body mass index (BMI) was depicted using spline curves, and univariate logistic regression analysis was performed using a restricted cubic spline model. A nonlinear cubic spline curve was drawn with the estimated probability and 95% confidence intervals for the outcomes. BMI: body mass index; MV: mechanical ventilation; PICS: post-intensive care syndrome.

**Table 1 jcm-11-05758-t001:** Patient characteristics and clinical course in first PICS survey.

	First PICS Survey (5.5 ± 3.1 Months after ICU Discharge)		
	PICS (*n* = 147)	Non-PICS (*n* = 104)	*p* Value
Age, yr, median (IQR)	68 (60, 75)	65 (56.3, 72)	0.010
Male, *n* (%)	114 (77.6)	86 (82.7)	0.32
BMI, kg/m^2^, median (IQR)	24.7 (22.0, 28.4)	25.9 (23.3, 29.0)	0.040
SOFA score on the day of ventilation start, median (IQR)	5 (4, 7)	5 (4, 7)	0.73
Clinical frailty scale before hospitalization, median (IQR)	2 (1, 3)	2 (1, 2)	0.27
Delirium, *n* (%)	36 (24.5)	14 (13.5)	0.030
Duration of delirium within 1 week of ICU admission, day, median (IQR)	2.5 (1, 5)	2 (1, 3)	0.30
Duration of invasive mechanical ventilation, day, median (IQR)	10 (6, 17)	8 (6, 13)	0.0030
Length of ICU stay, day, median (IQR)	13 (8, 21)	10 (8, 17)	0.010
Length of hospital stay, day, median (IQR)	26 (15, 51)	20 (10, 32)	0.0030
Comorbidity, *n* (%)			
Hypertension	62 (42.2)	52 (50)	0.22
Diabetes	49 (33.3)	31 (29.8)	0.56
Cardiac disease	13 (8.8)	13 (12.5)	0.35
Chronic kidney disease	3 (2.0)	4 (3.9)	0.32
Autoimmune diseases	7 (4.8)	2 (1.9)	0.20
Malignant tumors	7 (4.8)	7 (6.7)	0.50
COPD	12 (8.2)	9 (8.7)	0.89
Immunodeficiency	5 (3.4)	3 (2.9)	0.56
Treatment received during hospital stay			
Reintubation, *n* (%)	9 (6.1)	1 (1.0)	0.040
ECMO, *n* (%)	17 (11.6)	16 (15.4)	0.38
Duration of ECMO, day, median (IQR)	14 (9, 18)	10.5 (9, 17)	0.79
Tracheostomy, *n* (%)	35 (23.8)	16 (15.4)	0.10
Corticosteroid, *n* (%)	118 (80.3)	76 (73.1)	0.18
Maximum prednisolone dose, mg/day, median (IQR)	44 (30, 100)	40 (0, 75)	0.030
Continuous neuromuscular blocking agent, *n* (%)	63 (42.9)	49 (47.1)	0.50
Prone position, *n* (%)	82 (55.8)	50 (48.1)	0.23
Continuous renal replacement therapy, *n* (%)	12 (8.2)	7 (6.7)	0.67
Rehabilitation program, *n* (%)	83 (56.5)	57 (54.8)	0.80
Time from ICU admission to rehabilitation program initiation, day, median (IQR)	5 (2, 20)	4 (2.5, 12)	0.37

BMI: body mass index, COPD: chronic obstructive pulmonary disease, ECMO: extracorporeal membrane oxygenation, ICU: Intensive Care Unit, IQR: interquartile range, PICS: post-intensive care syndrome, SOFA: Sequential Organ Failure Assessment.

**Table 2 jcm-11-05758-t002:** Patient characteristics and clinical course in second PICS survey.

	Second PICS Survey (13.5 ± 3.2 Months after ICU Discharge)		
	PICS (*n* = 127)	Non-PICS (*n* = 82)	*p* Value
Age, yr, median (IQR)	68 (60, 75)	66 (56, 73.3)	0.12
Male, *n* (%)	98 (77.2)	69 (84.1)	0.22
BMI, kg/m^2^, median (IQR)	24.7 (22, 27.8)	25.9 (23.1, 29.1)	0.049
SOFA score on the day of ventilation start, median (IQR)	5 (4, 7)	4.5 (3, 7)	0.26
Clinical frailty scale before hospitalization, median (IQR)	2 (1, 3)	1 (1, 2)	0.26
Delirium, *n* (%)	25 (19.7)	13 (15.9)	0.48
Duration of delirium within 1 week of ICU admission, day, median (IQR)	2 (1, 4)	2 (2, 4)	0.58
Duration of invasive mechanical ventilation, day, median (IQR)	9 (6, 17)	9 (6, 14)	0.53
Length of ICU stay, day, median (IQR)	11 (8, 21)	11 (8, 17)	0.39
Length of hospital stay, day, median (IQR)	23 (14, 43)	20 (11, 35)	0.13
Comorbidity, *n* (%)			
Hypertension	56 (44.1)	40 (48.8)	0.51
Diabetes	32 (25.2)	29 (35.4)	0.11
Cardiac disease	9 (7.1)	9 (11)	0.33
Chronic kidney disease	4 (3.1)	2 (2.4)	0.56
Autoimmune diseases	5 (3.9)	2 (2.4)	0.44
Malignant tumors	6 (4.7)	4 (4.9)	0.60
COPD	10 (7.9)	7 (8.5)	0.86
Immunodeficiency	4 (3.1)	1 (1.2)	0.35
Treatment received during hospital stay			
Reintubation, *n* (%)	7 (5.5)	2 (2.4)	0.24
ECMO, *n* (%)	12 (9.4)	13 (15.9)	0.16
Duration of ECMO, day, median (IQR)	14 (8.3, 23.3)	10 (9, 14.5)	0.57
Tracheostomy, *n* (%)	24 (18.9)	15 (18.3)	0.91
Corticosteroid, *n* (%)	102 (80.3)	62 (75.6)	0.42
Maximum prednisolone dose, mg/day, median (IQR)	44 (30, 100)	42.6 (15, 82.5)	0.20
Continuous neuromuscular blocking agent, *n* (%)	54 (42.5)	39 (47.6)	0.47
Prone position, *n* (%)	70 (55.1)	43 (52.4)	0.70
Continuous renal replacement therapy, *n* (%)	13 (10.2)	5 (6.1)	0.30
Rehabilitation program, *n* (%)	72 (56.7)	42 (51.2)	0.44
Time from ICU admission to rehabilitation program initiation, day, median (IQR)	6 (2, 16)	4 (2, 19.3)	0.60

BMI: body mass index, COPD: chronic obstructive pulmonary disease, ECMO: extracorporeal membrane oxygenation, ICU: Intensive Care Unit, IQR: interquartile range, PICS: post-intensive care syndrome, SOFA: Sequential Organ Failure Assessment.

**Table 3 jcm-11-05758-t003:** PICS outcomes; assessment of PICS in first and second survey.

	First PICS Survey Assessment of PICS (*n* = 251)	Second PICS Survey Assessment of PICS (*n* = 209)
Dyspnea, *n* (%)	118 (47.0)	96 (45.9)
Walking difficulty, *n* (%)	89 (35.7)	54 (25.8)
Weight loss, *n* (%)	154 (61.4)	48 (23.0)
Memory impairment, *n* (%)	74 (29.7)	66 (31.6)
Executive dysfunction, *n* (%)	120 (47.8)	93 (44.5)
Depression, *n* (%)	103 (41.0)	81 (38.8)
Anxiety, *n* (%)	144 (57.4)	107 (51.2)
Sleeping disorder, *n* (%)	113 (45.0)	92 (44.0)
Visual analog scale, median (IQR)		
Physical condition (on a scale of 1 to 10)	7.3 (5.5, 8.5)	7.4 (6.2, 8.7)
Cognitive function (on a scale of 1 to 10)	9 (7.4, 9.9)	8.6 (7.1, 9.9)
Mental health (on a scale of 1 to 10)	8.3 (6.0, 9.5)	8 (6.5, 9.4)
Barthel Index, median (IQR)	100 (95, 100)	100 (95, 100)
Short-Memory Questionnaire, median (IQR)	40 (34.8, 44)	39 (34, 43)
HADS score, median (IQR)	8 (3.8, 14)	7 (3, 14)
HADS-Anxiety score	4 (1, 7)	3 (1, 7)
HADS-Depression score	4 (1, 7)	4 (1, 7)
EQ-5D-5L, median (IQR)	0.831 (0.710, 1)	0.844 (0.759, 1)

HADS: Hospital Anxiety and Depression Scale, IQR: interquartile range, PICS: post-intensive care syndrome.

## Data Availability

Individual participant data that underlie the results reported in this article are available from the corresponding author upon reasonable request.

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
