# Peer review of "Prevalence and Risk Factor Analysis of Post-Intensive Care Syndrome in Patients with COVID-19 Requiring Mechanical Ventilation: A Multicenter Prospective Observational Study"

_jcm, 2022, doi:10.3390/jcm11195758_

Round 1
Reviewer 1 Report
Dear authors,
The COVID-19 pandemic has led to a parallel pandemic of post-ICU complications, such as PICS. I read your manuscript and i found it well-written. I have some concerns about the interest of the study to the readers of the journal.
Overall comments: Well written manuscript. Good english language style and grammar use.
Abstract: It summarises all the important results of the study.
Introduction, Materials & Methods, Results: Well written. Maybe PICS criteria/definition could be described better.
Discussion, Conclusions: It would be interesting to comment on the differences between the first and second survey prevalence of PICS (58.6% vs 60.8%), as well as patient characteristics and outcomes.
Best regards
Reviewer 2 Report
The paper examined the prevalence and risk factors of post-intensive care syndrome (PICS) in ventilated patients with COVID-19 after ICU discharge. 60% of patients experienced persisted PICS, delirium and duration of mechanical ventilation were independent risk factors for PICS. Nowadays these aspects are challenge to care health system.
The paper covers an interesting topic and contributes to knowledge in the field. The paper is well written and organized. Figures and tables illustrate the study outline, patient characteristics and results. The introduction provides arguments to justify the study. The discussion is comprehensive.
Author Response
I would like to thank you for your review and appreciation of our manuscript.Reviewer 3 Report
The rational of the work is well presented and one easily understand its aim.
The abstract may be improved, structured, and include more results (cut-off, statistical significance...)
Reviewer 4 Report
The Paper introduction is not well written its too brief kindly increase the introduction.
THe previous literature is not written at all kindly include the table configuring the table including the same
Reviewer 5 Report
In this study, the authors report and discuss results from a Japanese multicenter study on the occurrence and risk factors of post-intensive care syndrome in Covid-19 survivors who received mechanical ventilation. The paper is well-written. The study contains an adequate sample size, statistical analysis, presentation and discussion, the references are up-to-date. Regarding the challenging question of defining and identifying PICS, the authors have found a good methodological approach with fitting outcome variables chosen as surrogate parameters for PICS components. The results are mostly in line with the current scientific literature on this topic. However, this study offers some novel insights in the form of detailed analyses of impairments in sub-groups and identification of some specific risk factors and associations in regression analyses.
There is only one minor point that should be addressed before publication:
1.) P4 L151: Please specify the primary outcome: How is the prevalence of PICS exactly defined for the primary outcome, e.g. any impairment (physical/cognitive/mental) in the respective questionnaires?
Reviewer 6 Report
Work very interesting, interesting, innovative.
Please adapt to the comments below:
- change of subject
- change of conclusions
- Figure 2. please make two separate figures unreadable
Reviewer 7 Report
This is a very interesting and large study on the prevalence of PICS in COVID-19 patients.
However, I have some major concerns.
1. How many of the 410 patients were previously hospitalised in the ICU?
How many of these 410 patients possibly had PICS due to their previous ICU admission?
I ask because you mention in your exclusion criteria that "patients who were unable to walk on their own before admission, regardless of the use of assistive devices, because of the possibility of pre-existing PICS" were excluded.
They could have however the cognitive/mental aspects of PICS (instead of the physical).
So it is strange that this is your only criterion for previous PICS exclusion.
2. Table 1: The p-value is for comparison within groups (eg PICS vs non-PICS at 5.5 months) and not between groups (eg PICS at 5.5 months vs PICS at 13 months?). Please present Table in another form, maybe split in two separate tables, to be more comprehendible what exactly you are comparing.
3. Table 1: You cannot have 12.8 or 16.8 days of stay or ventilation duration. Round to the nearest number.
4. Results: PICS was diagnosed in 147 patients (58.6%) in the first survey (with a lot of differences between PICS and no PICS as seen in Table 1), and 127 patients (60.8%) in the second survey (with the only difference between PICS and no PICS the BMI). How do you explain this discrepancy? I mean a higher percent of PICS, but no confounders.
5. Table 2: Missing p-values between first and second survey.
6. It confuses me however that as shown in Table 2 there are no apparent differences between 5.5 and 13 months. So why do you see those differences in Table 1? E.g. why is age statistically significant in 5.5 months and not 13 months? So my real objection is that in survey #2 you lost 20 PICS patients and 22 non PICS (?), or some had PICS at 5.5 months not at 13 months. You have to provide this information also, so we can compare the 2 surveys, or else it is a different population and no comparisons/conclusions can be drawn.
7. Conclusions: Delirium and longer mechanical ventilation were identified as independent risk factors for PICS.
Not true, only at 5.5 months, but not at 13 months.
8. I think you should only analyse those patients who answered both surveys (i.e. 209 patients), or else your methodology and results become confusing.
Round 2
Reviewer 1 Report
I find the corrections of the manuscript sufficient and I thank the authors for their reply.
Author Response
I would like to thank you for your review and appreciation of our manuscript.
Reviewer 7 Report
I thank the Authors for answering comprehendingly all my comments
Author Response
I thank the Authors for answering comprehendingly all my comments.